# What Can Laboratory Animal Facility Managers Do to Improve the Welfare of Laboratory Animals and Laboratory Animal Facility Staff? A German Perspective

**DOI:** 10.3390/ani14071136

**Published:** 2024-04-08

**Authors:** Birte von der Beck, Andreas Wissmann, Rene H. Tolba, Philip Dammann, Gero Hilken

**Affiliations:** 1Central Animal Laboratory, University Hospital Essen, University Duisburg-Essen, Hufelandstrasse 55, 45147 Essen, Germany; birte.vonderbeck@uk-essen.de (B.v.d.B.); andreas.wissmann@uk-essen.de (A.W.); 2Institute for Laboratory Animal Science & Experimental Surgery, Faculty of Medicine, RWTH Aachen University, 52062 Aachen, Germany

**Keywords:** animal welfare, human welfare, culture of care, compassion fatigue

## Abstract

**Simple Summary:**

Under the term *Culture of Care*, procedures have been developed to reduce the distress caused to animals and humans by animal experimentation. It has been recognized that animal welfare is human welfare and vice versa. Here, we look at the distress that laboratory animals and the staff are exposed to in animal experimentation. We also focus on the question of what laboratory animal facility managers can do to improve the welfare of laboratory animals, especially mice and other rodents, and the staff involved.

**Abstract:**

Laboratory animal facility managers must ensure that animal experiments can be carried out under optimal scientific conditions, that all legal requirements are met, and that animal welfare is maximized. Animal experimentation is stressful not only for the animals involved but also for the people who maintain these animals or carry out the experiments. Many of those involved find themselves in a constant conflict between scientific necessity, care, and harm. Under the term *Culture of Care*, procedures have been developed to reduce the burden of animal experimentation on the animals and the staff involved. The focus here is on what laboratory animal facility managers can do to improve the welfare of laboratory animals and the people working with them. Exemplary measures are the improvement of the housing conditions of laboratory animals, the introduction of uniform handling measures, clear and transparent structures via a quality management system, implementation of a no-blame culture of error (e.g., via Critical Incident Reporting System in Laboratory Animal Science [CIRS-LAS]), and open and respectful communication with all parties involved in animal experimentation, including the public and representatives of the authorities (public webpage, open house policy). The 6 Rs must be considered at all times: replacement, reduction, refinement, respect, responsibility, and reproducibility. We are writing this article from the perspective of laboratory animal facility managers in Germany.

## 1. Introduction

Many scientific breakthroughs have been achieved via the use of laboratory animals. These animals have helped human patients immensely and, therefore, deserve our gratitude and respect. Relevant parts of the public are critical of animal experimentation [1], although most experts consider it necessary in certain areas, such as basic or applied research [2,3,4]. By definition, animal experimentation induces pain, suffering, and harm to the animals [5], the cut-off level being the introduction of a needle. Animal experimentation, therefore, requires that society grant an exemption for interventions on animals that would otherwise be against the law. It is, therefore, justified that animal laboratories are under strict supervision by the authorities and the public. In most European countries, animal experiments require regulatory approval, which can be granted only if the scientific questions cannot be answered with alternative methods, such as cell cultures, organoids, organ-on-a-chip, or in silico methods. The evaluation of applications for animal experiments involves a balancing of interests, weighing the expected scientific progress against the expected distress to the animals.

In this paper, we use the term ‘animal facility management’ to include the head of the facility, his or her deputies, and the animal welfare officers. While in some establishments, these functions may be strictly separated, in others, they may be carried out by the same person. In Germany, these functions are usually carried out by veterinarians and, in rare cases, by biologists or doctors. Arrangements and roles vary from country to country. In Germany, animal facility managers have authority and a wide range of responsibilities. They have to provide the best possible care, taking into account the husbandry and experimental conditions in their facility. In addition, they must ensure that animal experiments can be carried out in their facilities under scientifically optimal conditions (state-of-the-art husbandry systems, surgery, and treatment rooms, well-trained facility scientists and animal technicians), that all legal requirements are met, and that the well-being of the animals under their supervision is ensured. In particular, they regulate the housing and handling of animals. In practice, animal facility managers and animal welfare officers must take into account the interests of the animals as well as those of many different stakeholders: scientists, animal technicians, regulatory authorities, and the public.

Societal attitudes towards animals have changed substantially in recent decades. Animals are no longer seen primarily as farm animals and helpers (e.g., in agriculture) or as a pure source of food but also as companions and family members [6]. The view that animals, as sentient beings, should be respected for their own sake has become established not only in morality but also in current law. The German Animal Welfare Act states in §1: “The purpose of this law is to protect the life and well-being of animals as fellow creatures from the responsibility of humans” [7]. As a result, so-called ethical animal protection is part of the constitution in Germany and the EU. The perceived intrinsic value of an individual animal has thus increased.

This change in perspective and value increases the psychological pressure on people who carry out animal experiments. People involved in animal experimentation almost inevitably experience psychological discomfort, if not crisis [8], throughout their professional lives because they are directly or indirectly involved in the suffering or killing of animals. Furthermore, although many medical breakthroughs have been made with the help of animal experiments [4], animal experiments are sometimes difficult to replicate, and their applicability to the development of specific treatments for human diseases is not always guaranteed. [9]. This problem can further increase the risk of psychological crisis for the people involved. It is now widely recognized that animal testing is distressing not only for the animals but also for the people who carry it out. According to Grimm [8], no fewer than nine of ten people in the profession will experience compassion fatigue at some point in their career.

The term *Culture of Care* describes a strategy aimed at developing policies that reduce the burden on people in caring professions [10]. Here, we follow this approach and apply it to the field of animal research (following other authors such as Williams [11]) and focus on the following question: What can laboratory animal facility managers do to improve the welfare of laboratory animals and laboratory animal facility staff? Specifically, we look at the stress to which animals and humans are exposed in animal experimentation and ask what strategies animal facility managers may use to reduce it. The underlying idea is that animal welfare promotes human welfare and vice versa.

## 2. Factors Affecting the Welfare of Laboratory Animals

Animal welfare encompasses many aspects, the most important of which are the absence of pain, physical and emotional well-being, and the ability to display the full repertoire of species-specific natural behaviors. In animal laboratories, all these factors are mainly influenced by the housing conditions, handling routines, experimental procedures, and, when appropriate, genetic modification and the age of the animals (Figure 1). Furthermore, individual and social factors can affect animal welfare as well, e.g., dominance behavior or pheromones. All these factors can interact. For example, the extent to which animals are stressed by experimental procedures may be modified by housing conditions. Additionally, wound healing in rodents is influenced by circadian rhythms [12,13]; therefore, it can make a big difference whether surgical procedures are scheduled during the active (dark) or resting (light) phase. In addition, animals in cages with environmental enrichment (refinement) recover more quickly than conspecifics housed in non-enriched cages [14,15]. It was shown that conventional laboratory housing increases morbidity and mortality in laboratory rodents [16]. Finally, the level of stress also depends on the expertise of the people conducting the experiments and whether the animals are familiar with these people or even with the procedures themselves. In some cases, physiology must be taken into account; for example, hungry rats are less sensitive to pain than rats that are not hungry [17].

### 2.1. Husbandry Conditions

Husbandry conditions include all factors of animal husbandry, including cage types and dimensions, ventilation of the animal husbandry rooms, day/night rhythm, bedding, food, water, climate, cage equipment, and social composition within the cages [18,19,20,21]. In European countries, housing conditions are required by law to meet the specific needs and characteristics of each species. This means that animals must be provided with shelter and food, and the environment in which they live and the food, water, and care they receive must be appropriate to their health, well-being, and species-specific requirements. Any restrictions on the extent to which an animal can satisfy its physiological and ethological needs should be kept to a minimum. Minimum standards for animal accommodation and care, including minimum cage dimensions, are set out in Directive 2010/63/EU, ANNEX III of the European Parliament and of the Council of 22 September 2010 on the protection of animals used for scientific purposes [2]. Deviations are permitted only for approved animal experimental procedures, such as short-term housing in metabolic cages.

Although the criteria for cage dimensions are clearly formulated, Directive 2010/63/EU is less clear on how laboratory animal cages should be equipped. It is clear that the equipment of the so-called “standard cage” in laboratory animal facilities has changed considerably over the past decades. Approximately 20 years ago, laboratory rodent cages were provided with little more than bedding material for the sake of standardization [22]. Nowadays, mice in most facilities are usually provided with at least nesting material and mouse houses as shelter. Very often, other items, such as tubes and gnawing sticks, are also added. Most animal facility managers and experimenters consider these changes to be beneficial for laboratory rodents without undue compromise of standardization efforts [23,24].

On the other hand, laboratory animal facilities have increasingly focused on efficiency and on housing as many animals as possible, especially mice and zebrafish, in response to the increasing number of genetically modified mouse and fish models. Their introduction has led to a massive increase in the number of animals in laboratory animal facilities. In the case of mice, maximizing the number of animals per room has led to a reduction in the average size of cages, particularly in individually ventilated cage (IVC) systems, which have largely replaced conventional open or filter-top cages. Today, IVC cages with floor areas of 500 to 540 cm^2^ for mice predominate in most modern animal laboratories. The cage racks are designed to accommodate a maximum number of cages per square meter.

This shift in cage systems has both advantages and disadvantages for the animals. Individual ventilation makes it easier to ensure the specific pathogen–free (SPF) status of the animals than in traditional open cage systems. It minimizes the risk of introducing unwanted pathogens, an improvement that is particularly relevant to the growing number of mouse strains with compromised immune function. On the other hand, a characteristic of IVC-based facilities is that the ventilation units generate a continuous air flow with associated background noise, which can lead to chronic stress in the animals [25,26]. Animals are completely isolated from their environment and are, therefore, unable to communicate with each other across cage boundaries by either odor or sound. This creates a dilemma for the management of species such as mice, which do not live solitarily but where males compete for access to females: in many cases, adult mice must be kept as singles to prevent injury or even death from fighting. Single-housing of male mice has been shown to be associated with various forms of distress, such as increased heart rate and disruption of the normal circadian sleep pattern [27] or increased anxiety and decreased exploratory behavior [28].

Furthermore, because of the small floor area of the cages, which is based exactly on the minimum legal requirements of the EU, the possibilities for environmental refinement [29] beyond the items listed above are limited in mouse IVC systems. In practice, there is barely enough space for a shelter mouse house, nesting material, and a tube, whereas the provision of other useful items, such as running wheels, is excluded, although laboratory mice (and even their wild ancestors [30]) consciously and extensively use them whenever they are available [31]. In the long run, these restrictions run the risk of chronically boring the mice.

### 2.2. Handling

Handling includes all interventions on the animals, such as changing cages (including cleaning intervals and the methods used to move mice from one cage to another), restraint (e.g., for health examination), weighing, application of substances, training, and transport. In the wild, because small rodents are exposed to many predators, their natural reaction to a larger, unfamiliar individual is to flee. In fact, all types of handling are, at least to some extent, stressful for mice [32], including the simple act of personnel entering the room [33]. When laboratory rodents are imported from outside breeders or facilities or even from one room to another within the same facility, it can take several days for stress-related behaviors and physiological parameters to return to baseline levels [34]. However, for obvious reasons, the handling of laboratory rodents cannot be completely avoided, as their health must be monitored regularly, bedding changed, cages cleaned, and experimental procedures performed.

In general, the welfare of laboratory animals depends crucially on the respect, qualifications, and skills of the people who treat or care for them. Recognizing deviations from normal behavior and correctly assessing signs of stress depends on the experience and expertise of the observer.

### 2.3. Experiments

In animal experimentation, the degree of stress, pain, or suffering inflicted on the animal depends on the interventions themselves (procedures, surgeries), the associated anesthesia and analgesia protocols, and their short- or long-term consequences. For example, the application of tumor cells may itself cause little stress and pain to the animal, but the growth of the tumor or the spread of metastases may later increase the stress to the animal.

### 2.4. Genetic Modification and Ageing

Today, most strains of mice kept and bred in animal laboratories are genetically modified. The breeding of mice that may, or are known to, develop a harmful phenotype caused by their genetic modifications is considered an animal experiment in Germany and the EU and is, therefore, subject to authorization [35].

The discussion about so-called “surplus animals” (laboratory animals that cannot be used for experimental purposes for various reasons [36,37]) has forced animal laboratories in Germany to establish “retirement homes” for mice that cannot be used in animal experiments because of their sex, age, genetics, or other reasons. The concept of retirement homes is to keep surplus animals for as long as possible, ideally until they die of natural causes. However, as in humans, older mice are susceptible to age-related diseases that do not occur in younger mice, and that can cause significant pain, harm, or distress. Defining humane and cumulative endpoints for aging laboratory animals is not a trivial task, and currently, no established standards are in place.

## 3. Factors Compromising the Welfare of People Involved in Animal Experimentation

### 3.1. Ethical Dilemma

From a humanistic point of view, scientific and medical progress has a positive connotation, whereas the opposite is true for causing pain and killing living beings. People involved in animal experimentation must certainly cause pain or death for animals, but whether this leads to scientific or medical progress is uncertain. As Ferrara [38] puts it: “As certain as the harm to laboratory animals is, as uncertain are the potential benefits to humans, animals or the environment”. In fact, most basic research findings will never be translated into concrete therapies for human patients, at least not in the short term. Even in those cases where they are, this translation usually takes several years.

This imbalance between the unavoidable psychological guilt on the one hand and the uncertainty and delay of the justifiable ‘pay-off’ on the other is a moral dilemma with a high potential for emotional distress. Such distress may be exacerbated by the so-called reproducibility crisis in science, which describes the observation that it is particularly difficult to reproduce the results of animal experiments from one laboratory in another [9,39]. That difficulty explains why three more principles have been added to the well-known 3 Rs (replacement, reduction, refinement): respect, responsibility, and reproducibility [40].

The animal–human relationship is determined, among other things, by the nature, professional context, and frequency of contact between animals and humans [41]. Although the dilemma described above applies to almost everyone involved in animal experimentation, the perspectives on animals and animal experimentation may differ between professional subgroups. For laboratory animal technicians, veterinarians, or zoologists (group 1), love of or interest in animals was usually the primary motivation for choosing their profession (Figure 2) [42]. Their interest in medical research and advancement has often been secondary, but they accept animal experimentation as unavoidable and necessary for this task. However, their premise remains that animals should suffer as little as possible. For medical doctors, medical students, and medical technicians (group 2), the welfare of human patients should be the primary motivation at the beginning of their professional education. However, other factors seem to be more important: interest in science, prestige, financial security, family background [43,44,45]. However, at some point of their scientific work, animal experiments may become necessary because they are the most appropriate “scientific tools” in medical research. The interest in animals and animal experimentation is therefore secondary. These differing primary interests can lead to conflicts: Group 1 is primarily concerned with the welfare of the animals and favours, for example, faster treatment or euthanasia if the animals are suffering. Group 2 is, at least according to our experience, often primarily interested in the successful completion oftheir experiments and therefore may favor, for example, the longest possible survival of the animals, even if they suffer. This divergence of interests can lead to conflicts which, if unresolved, can add to the emotional distress described above. Working with animals is challenging for people in both groups: They regularly see suffering but cannot always prevent it in time; they are directly or indirectly involved in the suffering or killing of animals and make decisions about life and death.

The killing of surplus animals is particularly stressful for animal technicians. The birth of such animals is unavoidable and fateful because they do not have the desired genotype or sex. Every laboratory animal technician must kill many unused mice every year. This act is particularly distressing because these animals provide no ‘scientific benefit’. Execution has an effect on the people who carry it out and can certainly lead to psychological crises. Working as an animal technician means causing harm to the same creatures you care for [8].

### 3.2. Working Conditions

Laboratory animal facilities are often located in basements, where climatic standardization is easier to achieve. By their nature, basements are not pleasant places to work, as people work under artificial light and air conditioning. Housing laboratory rodents in IVC racks is also physically and mentally demanding work for animal technicians. These facilities are a form of factory farming. Animal technicians work in a relatively monotonous environment, surrounded by IVC racks with a large number of cages containing many mice. This kind of monotony makes it difficult to concentrate, but as animal technicians are responsible for living creatures, any mistake can be fatal. Daily monitoring is much more difficult in this animal facility environment, and the animals lose their individual status.

The high-density cage racks are also not labor-friendly for the animal technicians who take care of the animals daily. According to the Directive 2010/63 EU, “Animals shall be checked at least daily by a competent person. These checks shall ensure that all sick or injured animals are identified and appropriate action is taken”. This check, therefore, requires the visual inspection of each IVC cage, an action that is not at all user-friendly because the highest and lowest rows of the IVC rack allow only impaired vision of the animals. Some animal houses use mirrors connected to an expansion stick so that the animals can be evaluated more easily.

### 3.3. Guilt and Compassion Fatigue

Mental health is a complex issue. The term *compassion fatigue*, sometimes referred to as the ‘negative cost of caring’, was originally introduced to describe the negative effects of regular exposure to patient emergencies experienced by hospital nurses [46]. Individuals working in other helping professions, including veterinarians and animal technicians, are also at risk of experiencing compassion fatigue [38,47,48]. Symptoms include difficulty concentrating, numbness or depression, social withdrawal, aches and pains, exhaustion, anger, or a reduced ability to feel empathy [49,50]. In line with these findings, the professional group of animal technicians has a strikingly high sickness rate, and some of them have mental health problems or other symptoms of compassion fatigue [8,40]. It cannot be ruled out that their daily work makes them sick and that they develop psychological crises based on compassion and empathy. Compassion is a human quality. It is useful and necessary in the field of animal experimentation! However, mental health is more complex than compassion fatigue, and not all mental health problems in animal facilities can be attributed to this. Some technicians reach states of burnout due to demotivation, lack of communication, personal conflicts, etc. The strategies for coping with each situation are different for each person. Facility managers can only provide conditions that minimize psychological stress. Serious mental health problems in laboratory animal facilities should be addressed and managed by psychologists.

### 3.4. Social Stigmatization

Animal technicians working in German laboratory animal facilities often state that they are reluctant to reveal their profession to others, e.g., at social events (own unpublished observations and personal communication). Scientists usually do not have this problem, but many of them also avoid talking about their decision to perform animal experiments or to kill animals for their research. Similarly, university public relations departments often advise their researchers not to mention animal experiments in press releases for new publications (personal experience). These reactions can be seen as indicators of an ambivalent social climate towards animal experimentation: majorities in most Western societies accept animal experimentation as necessary for scientific and medical progress but still stigmatize it and those who perform it as cruel.

## 4. What Can We Do to Improve the Situation for the Animals?

### 4.1. Husbandry Conditions and Handling

In particular, the animal facility manager and veterinarian, as well as the animal welfare officer, have a regulatory influence on the facility and on the treatment of the animals. Providing the best possible care for their animals is the primary responsibility of animal facility managers, partly out of moral responsibility for the animals [51] and partly because good science depends on good animal husbandry [40,51]. In the EU, detailed regulations are in place to ensure that laboratory animals are generally kept in appropriate conditions, particularly in terms of space. However, it should be noted that the legislation has defined only the minimum requirements for each species in Directive 2010/63/EU. Therefore, for animal facility managers, there is a potential for improvement in almost every aspect. In general, because animal facility managers are obliged to optimize the welfare of the animals under their responsibility, it is their responsibility to minimize the burden of the housing conditions on the animals, e.g., by providing appropriate cage equipment (Figure 3; environmental enrichment as refinement—in coordination with those responsible for the specific animal experiment) and sufficient space (Figure 4). As outlined above, the most widely used cage systems currently have floor areas that are based on the exact minimum of the EU legal requirements, which limits the possibilities of providing environmental enrichment items. It is clear that not every enrichment item on the market is granted by the animals, so the items offered should be validated before use and should be chosen carefully [29] in the aspects of altering research results and benefits to the animals. However, it is also obvious that some items, such as running wheels, whose positive effects on laboratory welfare are well documented [30], are not part of most standard cages, mainly because of space limitations or standardization concerns. In other words, larger cages would be beneficial for mice because larger cages would allow more and better environmental refinements for their inhabitants. The need to provide animals with more floor space is also supported in terms of hygiene and welfare [52]. Fuochi et al. 2023 clearly mention the need to review the minimum cage space requirements described in EU legislation. However, regulatory changes in this direction would come at a substantial cost: existing cage systems would have to be replaced on a large scale, and maintenance capacity per facility would decrease. However, if animals are to be housed in the best way possible, constructive discussions about space requirements are needed.

As long as space is the most limiting factor, varying the use of environmental enrichment devices over time (i.e., providing different items on different days or weeks) can be a valuable approach to preventing boredom in laboratory animals [53]. This approach is widely used in neuroscience but has been surprisingly little discussed and studied in laboratory animal science. Of course, such an approach is time-consuming and, to some extent, a disruptive factor for standardization [54,55]. Nevertheless, the strict interpretation of standardization promoted a few decades ago has been criticized by many authors [9,56] and is now considered outdated by most modern facility managers. A plethora of publications have demonstrated the negative influence of deprived facility conditions and the benefits of more stimulating environments on the well-being of laboratory animals [57,58]. Figure 3 shows the development of the standard mouse cage in the Central Animal Laboratory Essen from 1995 to 2021. It is incumbent upon the community to discuss, develop, validate, and implement further ideas in this area.

As described above, the cage areas offered are based on the minimum required by the European Directive. It is undisputed that larger cage areas would be beneficial for the animals. Even if it is not possible to change the cage systems for the whole population, such a change could at least be considered for certain subgroups, e.g., retired mice or mice undergoing handling training (see Figure 5).

In the area of handling, important steps have already been taken to reduce animal distress. For example, cage change intervals have been re-evaluated in recent years after telemetry studies had shown that too frequent transfers to new cages were stressful for the rodents, mainly because the new cages did not have the original odor of the previous ones [59]. In the age of the IVC rack, whose ventilation units remove all harmful environmental toxins (e.g., ammonia and CO_2_), frequent changes (1–2 times per week) are no longer recommended, and intervals have been increased accordingly.

Other aspects of daily handling are also important for animal welfare. For example, the German Animal Welfare Act, based on the EU Directive 2010/63, obliges facility managers to check the health status of each animal daily, regardless of whether it is part of an experimental procedure. These daily health checks must be carried out in a way that causes as little distress as possible to the animals. Additionally, if possible, animals should be handled by the same group of animal technicians throughout their lives; frequent changes of personnel should be avoided. This recommendation also applies to experimenters.

Modern facilities often house thousands of laboratory animals; consequently, a large number of people handle them. It is, therefore, important to establish standardized handling routines in any animal facility. Improper handling can become a negative social experience and can lead to a measurable stress response in laboratory animals [32]. Handling procedures should be standardized at all stages of the procedure, from the animal technician and veterinarian to the experimenter. This uniformity can and should be achieved by requiring all involved to attend hands-on handling courses and to follow standard operating procedures.

Some technical aspects of facility conditions may also play a role. Because most environmental factors, such as air temperature and humidity, are kept constant and vary little between facilities, the most important variable is probably the light-dark rhythm in the animal rooms. Most laboratory rodents are nocturnal and would benefit from a reversed light rhythm to synchronize their handling and treatments with their own active phases of the day. However, this would mean that animal technicians and experimenters would have to work in near darkness (via red light) throughout their working day, which is problematic from a humane welfare perspective. Therefore, animal facility managers may wish to consider the implementation of inverted light regimens for selected groups of animals or of experiments that are particularly sensitive to biological rhythms and activity periods, e.g., animals used in behavioral experiments, endocrinological studies, or surgeries that cause severe suffering and long recovery periods. However, these decisions do not rest solely with the animal facility manager; they must be made in consultation with those responsible for the specific animal experiment.

### 4.2. Experiments

Because of the invasive nature of animal experimentation, it is incumbent upon all involved to minimize the distress to the animals and to preserve their welfare as much as possible.

In the case of experiments, animal welfare officers must ensure that the experiments are carried out in accordance with the license and that the stress on the animals does not exceed the approved severity level. As the proposals for animal experiments have been reviewed and approved by the authorities from a wide range of perspectives, the management of the animal house does not have much influence on the experiments once they have been approved. Management and animal welfare officers can and must only ensure that animal experiments are carried out with the authorities’ approval. However, Animal Welfare Officers have a role to play in ensuring that the best practices are implemented in applications for animal experiments before they are submitted for approval. If various procedures prove to be unsuitable in the experiment, this must be recognized and corrected. However, all changes must be reported to and approved by authorities beforehand. Therefore, animal welfare officers must have excellent expertise and must ensure that this expertise is kept up to date with the latest developments in laboratory animal science. Similarly, the institutions in which they are employed must enable them to keep up to date with the latest research and animal welfare developments (e.g., allow sufficient time for congresses, training, courses, etc., to fulfill the obligation for continuous professional development).

Similarly, it must be ensured that all experimenters have the appropriate expertise in animal handling and that they keep up to date regarding scientific advances and refinements. When complex surgical techniques are used in working groups, care should be taken to ensure that only a small number of specialists are involved in the surgery. This will prevent animals from suffering unnecessarily if such procedures are carried out by inexperienced researchers (such as postgraduate students).

Early detection of distress caused by the experimental interventions is essential and must be included in the experimental protocols, e.g., using scoring sheets and documentation of all the relevant and procedure-tailored symptoms of distress. Whenever possible, distress should be reduced (refinements such as analgesia, soft bedding, softened diet, cage ground level, etc.

In recent years, medical training has become increasingly popular. Medical training is the teaching of specific behaviors and exercises to assist in medical treatment, examination, or care. Training prepares the animal for unfamiliar situations and, if designed appropriately (usually with the provision of rewards), can even lead to a positive association over time. Thanks to previous training, the animals know exactly what to expect and, therefore, experience significantly less stress and anxiety. This concept has long been common practice for large laboratory animals (e.g., dogs and pigs), but it is only slowly becoming established for small laboratory rodents [60]. The incredible successes that can be achieved can be seen, for example, in the impressive videos from the Swedish Research Institute [61]. However, when it comes to medical training, it is important to consider when such training can be a disruptive factor and when it is definitely beneficial for the animals. Specific medical training does not make sense for short-term experiments in which the animals are handled only a few times. Then, the burden of medical training is likely to predominate. In the case of longer experiments, in which the animals must be manipulated repeatedly, the benefit to the animals can be immense and therefore justifies a considerable additional effort for the people involved. This benefits not only the animals but also the quality of the scientific results [40,51].

Medical training can also be applied in the context of transport. Transport should be carried out in such a way as to minimize distress to the animals. Before demanding experiments, such as stressful surgical procedures, transport to other rooms can be trained before the actual procedures are performed. In this case, the transport does not add to the stress of the procedure because the animals are already accustomed to it. The longer the distance traveled, the more time should be allowed for the animals to recover in the treatment room before any procedures are performed. Because medical training can be time-consuming, experimenters must work hand in hand with animal facility staff.

## 5. What Can We Do to Improve the Situation for the People Involved in Animal Experimentation?

As described above, keeping and experimenting on laboratory animals involves an ethical dilemma that can, in the long run, cause severe emotional stress and even “compassion fatigue” in many of the people entrusted with it [38].

There is no way to escape this dilemma completely. Therefore, the management of an animal facility must reassure all those involved that they are working in a facility that promotes the welfare of the animals to the maximum while at the same time minimizing the negative aspects of animal experimentation. In concrete terms, this means doing everything possible to provide an environment that meets the needs of the animals and promotes their well-being. Such actions require constant critical self-examination and, if necessary, adjustment of routines and practices when they can be improved (Figure 5). A good example of this is cage size and equipment; another is the way in which small laboratory rodents are moved from cage to cage. It has recently been shown that animals benefit significantly from being moved by tunnel tubes when changing cages rather than by tail restraint [62]. It is, therefore, up to the animal facility manager to keep abreast of such developments and, when necessary and possible, to implement them promptly and decisively in their own facility.

An important step is to cooperate with the institution’s occupational health and physiotherapy services, which can help introduce mental rest tools and active break management or teach staff how to take care of the body during repetitive movements. Another very important aspect is the communication culture (Figure 3). Open and effective communication between all those involved in an animal laboratory is essential. It is equally important that innovations and adjustments are not decided and implemented by management alone but that scientists and animal technicians are involved in the decision-making process. It is primarily the animal technicians, the veterinarians, and the experimenters who are in direct contact with the animals, who see their particular suffering, who must care for them and, if necessary, euthanize them. This is one of the reasons that it is so important to convince animal technicians of the benefits of a proposed innovation from the outset or, if necessary, to give them the opportunity to dissuade management. Such a culture of communication between animal house management and animal technicians may be more strenuous than a classic top-down approach. This approach has the potential to increase the psychological resilience of animal technicians, who are particularly vulnerable to compassion fatigue, by making them feel that they are active players in the overall process rather than just assistants or soldiers who simply receive orders. The working atmosphere and communication culture should encourage animal technicians to communicate any undesirable developments or grievances they perceive in their area of responsibility without fear of sanction.

Equally important is constant and constructive communication between the animal facility managers and the research staff. Scientists should work in the knowledge that the entire staff of the animal facility will support them in their research to the best of their ability. It is essential that everyone involved, including the animal technicians, is confident that the experiments being carried out in the facility are always ‘state of the art’. For this reason, we believe it is necessary to work according to uniform and pre-defined rules, e.g., a quality management system. This adherence to guidelines ensures that procedures and actions comply with the law and animal welfare standards; it also guarantees safety in this sensitive area. It is also necessary to develop and maintain an error culture that discourages the repetition of unnecessary errors. Therefore, animal facility managers and animal welfare officers should not limit themselves to helping researchers apply for animal projects and then controlling them. Rather, both groups should see it as their responsibility to keep abreast of the latest developments in animal welfare and to apply this knowledge to their own projects without delay.

Constructive communication between researchers and animal technicians must also be encouraged. When researchers regularly update animal technicians on the aims of their projects and the progress made, this counteracts the fatal disconnect between the observed (or even self-inflicted) suffering of the animals and the ultimate purpose that should justify it. It has proved very useful for researchers to give lectures to animal facility staff, discussing their research and reflecting on the results. Such interaction strengthens networking and links between professional groups.

Communication is a key factor in reducing the risk of compassion fatigue. Open and intensive communication is necessary between all those involved in an animal facility and in an experiment. Communication must be at eye level and respectful. Addressing the scientific question is the key to justifying animal experimentation. Patients should be helped effectively! Finally, we believe that it is important for animal technicians not to hide their work from the public but rather to report honestly, reflectively, and self-confidently about their tasks and their work. The skepticism of the authorities and the concerned public can be countered only by an open house and a transparent communication culture. Events and guided tours must give the public the opportunity to convince themselves of the need for animal testing. In the long term, this openness may be the best way to counteract the social stigma against the use of animals for research purposes and social disregard for all the professionals involved.

## 6. Conclusions and Outlook

We have written this article from the perspective of German animal facility managers. Many scientific breakthroughs have been achieved via the use of laboratory animals. These living beings, therefore, deserve our gratitude and our utmost care. The ethical dilemma surrounding the use of animals in biomedical research can lead to psychological discomfort and sometimes crises of compassion among staff. Compassion is a human quality—as such, it is natural, meaningful, and necessary. The implementation of a culture of care aims to reduce the burden on animals and humans in animal experimentation.

It is essential to prepare animal experiments as well as possible. The 6Rs—replacement, reduction, refinement, respect, responsibility, and reproducibility—must always be considered. We must keep animals well, treat them with love and professionalism, avoid unnecessary suffering, and provide adequate medical care. Experimenters must constantly optimize handling and experimental techniques and breed their mouse strains with precision to avoid surplus animals (breeding management).

Addressing current scientific questions appropriately is the key to justifying animal experimentation. Open and effective communication in an experiment is necessary to make the meaning of each experiment clear to all participants. In medical science, the ultimate goal of research is to help patients effectively. Upcoming skepticism of the authorities and the concerned public can only be countered by an open house culture. Events and guided tours can give the public the opportunity to convince themselves of the necessity of animal experiments.

Last but not least, working according to the guidelines of a quality management system and developing and maintaining an error culture are further important steps to learn from mistakes or avoid repeating them.

## Figures and Tables

**Figure 1 animals-14-01136-f001:**
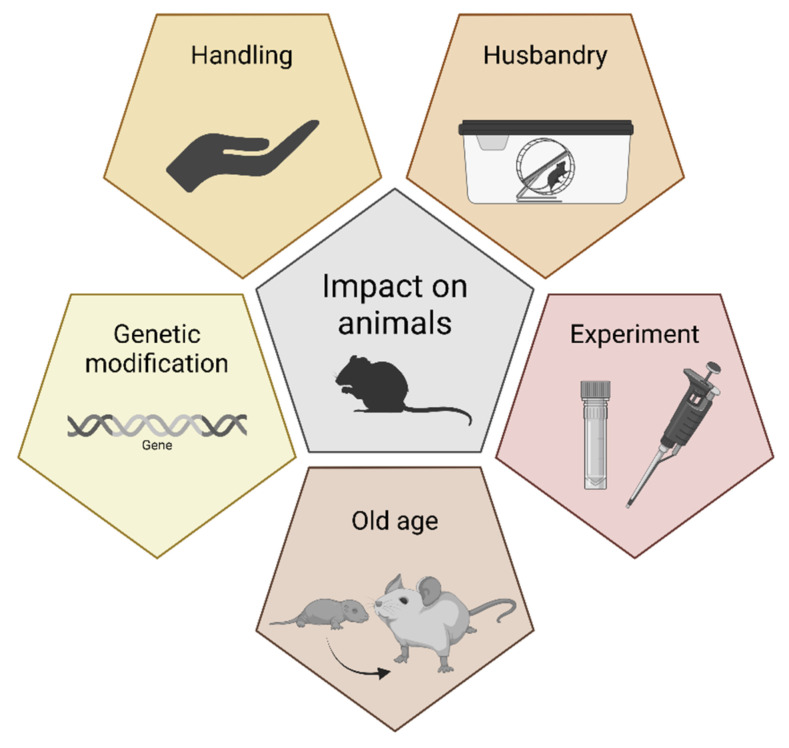
Factors affecting the welfare of laboratory animals.

**Figure 2 animals-14-01136-f002:**
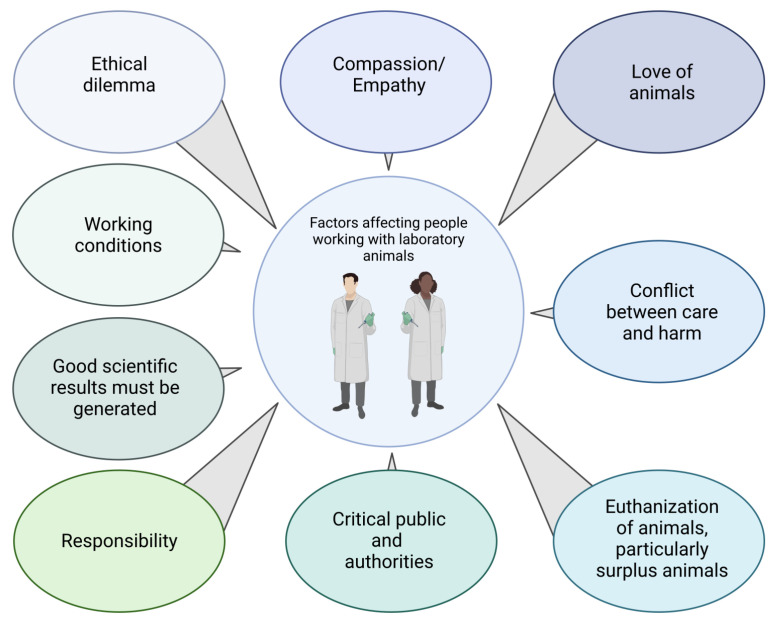
Factors affecting people working with laboratory animals.

**Figure 3 animals-14-01136-f003:**
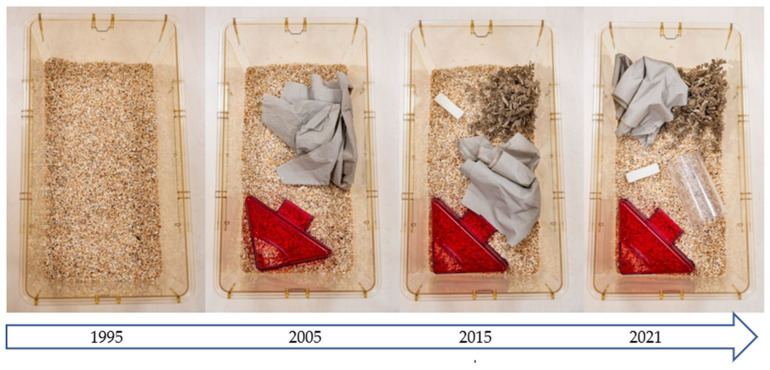
The development of the standard mouse cage in Central Animal Laboratory Essen from 1995 to 2021.

**Figure 4 animals-14-01136-f004:**
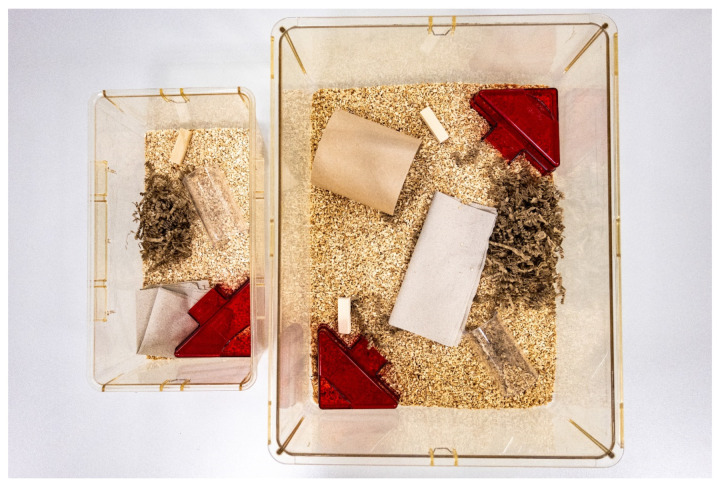
Standard cage (**left**) vs. cage for “retired” mice or mice used in, e.g., training programs (**right**).

**Figure 5 animals-14-01136-f005:**
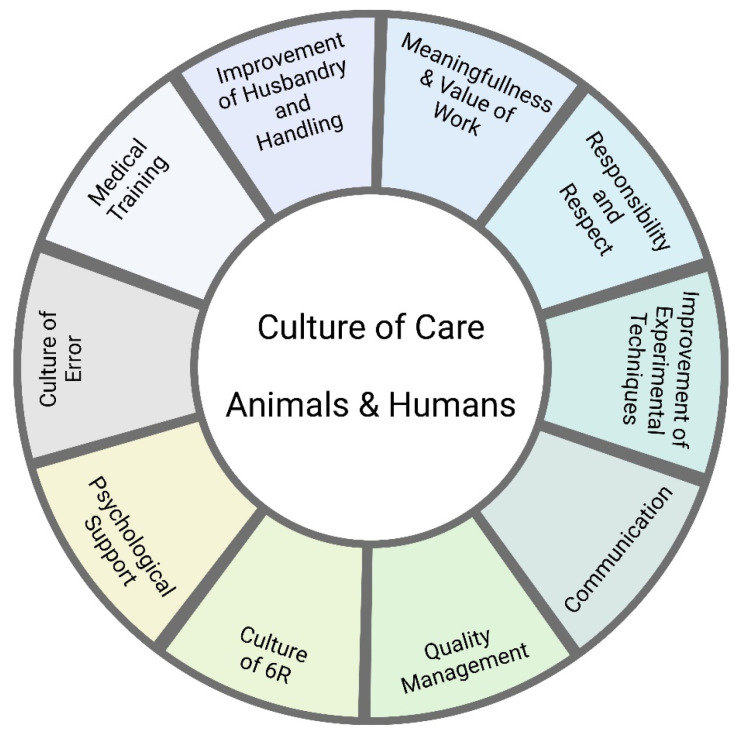
Synthesis: What can we do to improve the situation for the animals and humans in laboratory animal facilities?

## Data Availability

Data are contained within the article.

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
