# Peer review of "What Can Laboratory Animal Facility Managers Do to Improve the Welfare of Laboratory Animals and Laboratory Animal Facility Staff? A German Perspective"

_animals, 2024, doi:10.3390/ani14071136_

Round 1
Reviewer 1 Report
Comments and Suggestions for Authors
The manuscript "What can Laboratory Animal Facility managers do to improve the welfare of laboratory animals and laboratory animal facility staff" addresses the importance of communication between stakeholders in animal facilities. It evidences the stressful factors behind animals and humans being exposed to animal experiments and formulates strategies for improving animal and human well-being.
The authors know the area and have plenty of experience in laboratory animal studies. They wrote the manuscript very well. The topic is original and possesses high current importance; although the manuscript is not as impactful as the title suggests. Indeed, the 15-page manuscript does not provide new information or new ways of addressing the problem of animal and human well-being in animal facilities. In that way, a shortening is highly desirable. My review is in the attached file.
Finally, I invite the authors to acknowledge explicitly the study's limitations and the next steps in that research direction.

Comments on the Quality of English LanguageAuthor Response
Please see the attachment

Reviewer 2 Report
Comments and Suggestions for Authors
The proposals in this paper are fine reminders of ways to improve animal welfare in laboratory animal facilities. If there is anything truly new here, I have missed it. Thus, while I believe this paper is both factually correct and ethically sound, I don’t see a reason for publishing it. What does it add?
Here’s my suggestion:
Consider that arrangements and roles differ in different countries, and that your paper might be valuable specifically within the German context.
Please start by defining the role of the animal facility manager and the animal welfare officer – are they different people at institutions? Sometimes one person fills both roles?
Then consider whether there are ways in which the Animal Facility Manager has a unique role and unique opportunities to promote human and animal welfare --- opportunities and authorities that the vets, ethics committee members, scientists, government regulators, and animal welfare officers may not have.
If there are none, why single out the AFM to be the subject of this manuscript?
Consider making an argument that facility managers SHOULD have more authority --- perhaps on the ethics committee -- because of the important perspective their job gives them
I do find this interesting, as I best know the US system, where only in rare circumstances in small institutions would an Animal Facility Manager have the knowledge an authority for the roles you describe in this paper.
Reviewer 3 Report
Comments and Suggestions for Authors
This manuscript renews the reader's concern for animal welfare. Both the points discussed in the article and the writing of the paper are excellent. I am more of a learner than a commentator.
I have only a few very small suggestions:
Line 35 Wouldn't it be better to replace "staff welfare" with "human welfare"? Looking at the keyword alone, "staff welfare" is not easy to understand.
Line 72 This sentence reads as overly dismissive of the role of animal experiments, and I think it needs to be revised to make the sentence more objective and accurate.
Line 193 The task must be enormous, as there will be a steady stream of "old mice". There is a paradox in balancing these breeding efforts with other human needs. In addition, will these mice continue to breed? And what happens if they produce offspring? This is very much like the "stray animal" problem, where shelters have limited capacity, but stray animals keep appearing.
As a whole, the manuscript focuses on summarising and analysing animal welfare issues in mouse experiments. However, animal welfare issues exist in any kind of animal-related research. In particular, birds, fish and invertebrates are also heavily used in animal experiments. I would suggest either adding a small number of analyses of other animals, or making it clear in the abstract that this paper is a discussion of mice.
In addition, the manuscript ignores the question of how to judge whether animal welfare is good. Scratching behaviour is a very common displacement behaviour in mice. There was a method developed by researchers (Current Zoology, 2022, https://doi.org/10.1093/cz/zoac035), the behavior random permutation model could “be used to detect differences in the behavior of individuals reared in isolation (stressful conditions) as opposed to individuals reared in an enriched environment (more akin to natural conditions)”. Maybe you could learn about this method.
Line 536 Was the title italicised by mistake?
Comments on the Quality of English LanguageMinor editing of English language required
Reviewer 4 Report
Comments and Suggestions for Authors
The Authors present a review on potential contribution of Facility Managers to improve the welfare of Lab animals and Animal care/facility staff. The review covers an initial general part, addressing factors impacting on welfare of both animals and staff, and a second part addressing potential solutions to mitigate distress for both animals and staff. Of note, Authors are all from the same EU Country (Germany), and the review has a clear EU and German perspective. This aspect should be mentioned in the title or as a remark to close the Introduction, as elsewhere also in Europe, responsibilities that are in this review described as part of Facility Mangers' tasks might be assigned to other Institutional roles (e.g. Animal Welfare Responsible or Designated Veterinarian).
Minor comments to address:
Lines 188-189: With regards to the sense of the sentence, it is more correct to state: The breeding of mice that risk to develop or are known to have a harmful phenotype caused by their genetic modification is considered an animal experiment... .
This, consistently with the Reference EU Document, to be cited accordingly: European Commission, Directorate-General for Environment, Framework for the genetically altered animals under Directive 2010/63/EU on the protection of animals used for scientific purposes, Publications Office of the European Union, 2022, https://data.europa.eu/doi/10.2779/499108
Lines 221-228: Assumptions regarding reasons for Group 1 vs Group 2 to begin their professional education provide no support references and can be questionable (at least for group 2). Please confront:
Consistently with Authors' assumption for Vet category from Group 1 - See Figure 28 of the FVE Report (2015):
Conversely to Authors' assumption for MD/Med Students from Group 2, factors like interest in science, prestige, financial security, family background seem to be predominant vs the welfare of humane patients:
Goel, S., Angeli, F., Dhirar, N. et al. What motivates medical students to select medical studies: a systematic literature review. BMC Med Educ 18, 16 (2018). https://doi.org/10.1186/s12909-018-1123-4
Watari, T., Nagai, N., Kono, K., & Onigata, K. (2022). Background factors associated with academic motivation for attending medical school immediately after admission in Japan: A single-center study. Journal of general and family medicine, 23(3), 164–171. https://doi.org/10.1002/jgf2.528
Narayanasamy, M., Ruban, A., & Sankaran, P. S. (2019). Factors influencing to study medicine: a survey of first-year medical students from India. Korean journal of medical education, 31(1), 61–71. https://doi.org/10.3946/kjme.2019.119
Suggestions for improvement:
Lines 42-43: in this case, Authors may consider to include in the sentence the cut-off level - that is the introduction of a needle - to underline the high level of precaution applied by the Law.
Lines 97-98: The positive effect of EE is not only valid to promote recovery, but in a wider perspective. Please consider referring here also to Cait et al, 2022: Cait, J., Cait, A., Scott, R.W. et al. Conventional laboratory housing increases morbidity and mortality in research rodents: results of a meta-analysis. BMC Biol 20, 15 (2022). https://doi.org/10.1186/s12915-021-01184-0
Lines 313-315: The need to provide animals with larger floor surface is also endorsed with regards to hygienic conditions and welfare. Authors may want to consider the paper by Fuochi et al, 2023 (cfr. Discussion), clearly mentioning the need to review minimal cage surface requirements described in EU legislation. Fuochi, S., Rigamonti, M., Raspa, M. et al. Data repurposing from digital home cage monitoring enlightens new perspectives on mouse motor behaviour and reduction principle. Sci Rep 13, 10851 (2023). https://doi.org/10.1038/s41598-023-37464-8
Lines 405-409: As published by one of the Authors of the Manuscript under review, body weight changes might be valid or controversial based on the specific animal model. I would avoid using this specific parameter as the only explicitly mentioned example in the sentence.
Please confront Talbot SR, Biernot S, Bleich A, et al. Defining body-weight reduction as a humane endpoint: a critical appraisal. Laboratory Animals. 2020;54(1):99-110. doi:10.1177/0023677219883319.
Authors may consider including more examples – and bibliographic references – or rephrasing in the following, more general form: Early detection of distress caused by the experimental interventions is essential and must be included in the experimental protocols, e.g., through scoring sheets and documentation of all the relevant and procedure-tailored symptoms of distress. Whenever possible, distress should be reduced (refinements such as analgesia, soft bedding, softened diet, cage ground level, etc.).
Lines 500-501: Social disregard affects multiple professions involved with in vivo research, starting from primary disapproval of animal experimentation itself. Authors may want to consider rephrasing as it follows, to make the closure remark more inclusive: In the long term, this openness may be the best way to counteract the social stigma against the use of animals for research purposes and social disregard for all the professionals involved.

Round 2
Reviewer 1 Report
Comments and Suggestions for Authors
Dear authors,
Accepted. I agree with your comments and fully agree with the overall improvement of the manuscript. The only comment I have is:
Lines 131 - 133: I did not suggest a change; I asked a question to understand the provided categorization of animal welfare. You are not welfarists, but the reflection is important.
I thank you for acknowledging that human mental health is complex. And I hope the facility managers involve human mental health professionals in discussing how to prevent, diagnose, address, and treat these common and critical problems.
Best Regards
Comments on the Quality of English LanguageOK.
Reviewer 2 Report
Comments and Suggestions for Authors
The authors have made several edits, most of them small and some of them more substantive. As a US-based reviewer, I was confused in the first draft because the title and role of Animal Facility Manager is frequently an animal care staff-person with little formal education in biology, who stays out of the way of the researchers and lets the vets and the ethics committee make important decisions about how animals should be housed, cared for, and manipulated.
By adding clarification and explication that this paper is rooted in the German system, it gives an international audience good insight into how senior animal facility managers (vets, scientists) can work to improve human and animal welfare.
I learned a lot from this paper, and recommend publishing it as is.